# Trimester-Specific Serum Lipid Profiles in Gestational Diabetes Mellitus: A Systematic Review, Meta-Analysis, and Meta-Regression

**DOI:** 10.3390/medicina61071290

**Published:** 2025-07-17

**Authors:** Milos Milincic, Andja Cirkovic, Katarina Ivanovic, Stefan Dugalic, Miroslava Gojnic Dugalic

**Affiliations:** 1Clinic for Gynecology and Obstetrics, University Clinical Centre of Serbia, 11000 Belgrade, Serbia; ikatarina.1996@gmail.com (K.I.); stef.dugalic@gmail.com (S.D.); miroslavagojnicdugalic@yahoo.com (M.G.D.); 2Institute for Medical Statistics and Informatics, Faculty of Medicine, University of Belgrade, 11000 Belgrade, Serbia; 3Faculty of Medicine, University of Belgrade, 11000 Belgrade, Serbia

**Keywords:** lipid profile, gestational diabetes mellitus, pregnancy, triglycerides, cholesterol

## Abstract

*Background and Objectives*: Gestational diabetes mellitus (GDM) is a major public health concern associated with adverse maternal and neonatal outcomes. It was found that even physiological pregnancy is followed by a significant shift in serum lipid profile, and even more pronounced in GDM pregnancies. We aimed to comprehensively assess lipid parameters among pregnant women with and without GDM. *Materials and Methods*: A systematic review, covering PubMed, WoS, and SCOPUS until 23 July 2024, with meta-analysis and meta-regression, was conducted, comprising studies measuring TG, TC, LDL-C, HDL-C, VLDL-C, and TG/HDL ratio in pregnant women diagnosed with GDM, and those with normal glucose tolerance. The overall effect size measure was the SMD. NOS and JADAD scales were used for quality assessment, I^2^ statistics for heterogeneity evaluation, and funnel plots for publication bias inspection. *Results*: A total of 457 studies were included in the qualitative analysis, and 74, 277, and 122 studies were included in the quantitative analysis for the 1st 2nd, and 3rd trimester, respectively. TG and TG/HDL levels were significantly elevated in all three trimesters (TG: SMD = 0.61, 0.57, and 0.48, *p* < 0.001 for all, and TG/HDL: SMD = 0.44, 0.66, and 0.49; *p* < 0.001 for all), while TC and LDL-C levels showed significant increases in the 1st and 2nd trimesters (TC: SMD = 0.38, 0.27, *p* < 0.001 for both, LDL-C: SMD = 0.33, 0.20, *p* < 0.001 for both), in pregnant women with GDM compared to those without the condition. *Conclusions*: GDM is associated with significant lipid abnormalities, particularly elevated TG and decreased HDL-C levels. These lipid changes are most pronounced in the first and second trimesters, highlighting the importance of early detection and management.

## 1. Introduction

Gestational diabetes mellitus (GDM), defined as the onset of glucose intolerance first identified during gestation, is a major metabolic disorder that arises during pregnancy [1]. GDM is diagnosed using the Oral Glucose Tolerance Test (OGTT) following various guidelines. Currently, the International Association of Diabetes and Pregnancy Study Groups (IADPSG) guidelines are the most commonly used. According to the IADPSG recommendations, GDM diagnosis is based on the 2-h 75-g OGTT, which is typically performed between 24 and 28 weeks of gestation [2].

GDM is a major global public health concern, with an estimated worldwide prevalence of approximately 14%. These rates range from 7.1% in North America and 7.8% in Europe to as high as 27.6% in the Middle East and North Africa [3]. GDM is the most common complication of pregnancy [4]. GDM is linked to various complications in both mothers and children, such as gestational hypertension, preeclampsia, increased risk of type 2 diabetes mellitus (T2DM), large for gestational age (LGA), macrosomia, and neonatal hypoglycemia [5,6]. Early detection and appropriate treatment of GDM are essential for reducing the risk of adverse outcomes. Proper timing of GDM treatment has been shown to reduce the incidence of macrosomia by more than 50% and that of shoulder dystocia by over 60% [7]. The American College of Obstetricians and Gynecologists (ACOG) and American Diabetes Association (ADA) recommend early screening for GDM in pregnant women with identified risk factors, including a body mass index (BMI) > 30, a positive family history of diabetes, a history of GDM in a previous pregnancy, polycystic ovary syndrome (PCOS), or a prior delivery of a macrosomic infant [8,9].

Optimal lipid metabolism during pregnancy is crucial for optimal fetal development, whereas dyslipidemia is linked to a range of adverse perinatal outcomes, affecting both short- and long-term maternal and neonatal health [10]. A physiological progressive rise in maternal circulating TG, TC, LDL-C, and HDL-C levels throughout pregnancy, primarily driven by estrogen, progesterone, human placental lactogen, and cortisol, reflects a maternal metabolic shift from lipogenesis in early pregnancy to lipolysis in later stages, ensuring a sufficient energy supply to the growing fetus [11]. TC levels increased by almost 50%, LDL-C by 30–40%, HDL-C by 25%, and TGs by two- to three-fold [12]. However, GDM represents the state of exaggerated and pathologically altered lipid metabolism adaptations, as a result of the synergistic effect of insulin resistance and hormonal changes, leading to a more pronounced dyslipidemic profile. Multiple studies have demonstrated that women with GDM have significantly elevated TG levels and reduced HDL-C concentrations compared to normoglycemic pregnant women [13,14]. These alterations are not transient but tend to persist throughout pregnancy, with increasing severity in the second and third trimesters, suggesting an underlying metabolic dysfunction beyond normal physiological adaptation [13,15]. Metabolic imbalance may not only facilitate the development of GDM but also increase the risk of the adverse outcomes mentioned above.

Given that lipid abnormalities are detectable early in pregnancy and tend to persist or worsen toward the time of GDM diagnosis, they may serve as early biomarkers for identifying women at an increased risk [16]. Recognizing women at increased risk for GDM in early pregnancy remains difficult, even though several well-known risk factors, such as maternal obesity, older maternal age, a prior diagnosis of GDM, and a positive family history of diabetes mellitus, have been identified [17]. Aging is associated with a progressive decline in insulin sensitivity, even in non-pregnant populations [18]. Smoking is a well-documented factor that increases TG levels and lowers HDL-C concentrations, contributing to a pro-atherogenic lipid profile [19]. Recent findings indicate that certain lipid biomarkers assessed in early pregnancy may act as early predictors of the subsequent onset of GDM [20,21]. Lipids, especially increased TG levels and reduced concentrations of HDL-C during early pregnancy, have been suggested as potential early indicators of GDM; however, the findings of various studies are inconsistent and highlight the need for further investigation [13,15]. Wang et al. (2015) indicated that women who developed GDM had significantly reduced HDL-C levels in early pregnancy, suggesting its role as a complementary predictive biomarker [22].

Disturbances in lipid metabolism identified in early pregnancy are increasingly recognized as promising indicators of the future onset of GDM [23]. Numerous observational studies have underscored the prognostic value of assessing maternal lipid profiles during the first trimester for early risk stratification [24,25]. Studies have also confirmed the maintenance of elevated lipid profile parameter levels throughout pregnancy, with a trend toward convergence in the third trimester. Therefore, this study aimed to systematically evaluate whether lipid profile parameters differ between women diagnosed with GDM and those with normal glucose tolerance. More importantly, we aimed to investigate whether these differences persisted at the time of GDM diagnosis, thereby supporting the hypothesis that dyslipidemia and GDM are interrelated metabolic disturbances. Additionally, this study sought to assess whether these lipid profile differences remained significant after accounting for established maternal risk factors, including pre-pregnancy obesity, advanced maternal age, smoking status, and other potentially influencing characteristics. These findings could contribute to early identification strategies and targeted interventions, potentially reducing the burden of GDM and its associated maternal and neonatal morbidities.

## 2. Materials and Methods

### 2.1. Study Design

The PRISMA protocol (Reporting Items for Systematic Reviews and Meta-Analyses) for conducting systematic reviews and meta-analyses [26] was applied within this previously PROSPERO registered research under the number CRD420251032919.

### 2.2. Eligibility Criteria

Original studies that assessed lipid profile parameters (TG, TC, LDL-C, HDL-C, VLDL-C, and TG/HDL ratio) in pregnant women with GDM compared to normoglycemic pregnant women were eligible for our systematic review.

The inclusion criteria were defined based on the PECOS system as follows: (P) type of population: pregnant women; (E) type of exposure: GDM; (C) type of controls: non-GDM; (O) type of outcomes: measured levels of TG, TC, LDL-C, HDL-C, VLDL-C, and TG/HDL ratio; (S) type of study design: controlled trials, prospective or retrospective cohort, nested case-control in cohort, case-control, and cross-sectional studies.

The exclusion criteria were as follows: (1) articles published in languages other than English; (2) non-original research articles, including narrative reviews, systematic reviews, meta-analyses, case reports, case series, editorials, commentaries, correspondences, books, book chapters, letters to editor, short, abstracts, etc.; (3) wrong population: non-pregnant women, pregnant women with a disease other than GDM, and non-human population (animals, cell lines); (4) no control group at all or inappropriate control group; (5) wrong outcome: lipid profile parameters other than TG, TC, LDL-C, HDL-C, VLDL-C, and TG/HDL-C ratio evaluated. In cases of cohort overlap, only the study that encompassed the other sample was included.

### 2.3. Search Strategy

The search strategy was developed and performed by two researchers with high competence in conducting systematic reviews and meta-analyses (AC and MGD). It was applied to three electronic databases: PubMed, Web of Science (WoS), and SCOPUS until 23 July 2024. The search queries used in these databases are presented in Table 1. Additionally, the reference lists of articles retrieved through electronic searches, as well as those of relevant reviews and editorials, were manually screened to identify additional potentially relevant studies.

### 2.4. Article Screening and Selection

Two independent reviewers (MM, KI) screened publications for potential inclusion in the systematic review in two phases: first, by reviewing the title and abstract, and second, by reading the full article. All conflicts were addressed through discussion at each stage with input from a third reviewer (AC). If a reviewer considered a publication potentially eligible or if there was insufficient information in the title and abstract to justify exclusion, the publication was included.

### 2.5. Data Abstraction and Quality Assessment

Two reviewers (MM, KI) individually reviewed and extracted the following data sorted into three sets: (1) study characteristics: author, publication date, country, and study design; (2) characteristics of cases and controls: type of DM, applied definition and criteria for DM, sample size, specific characteristics of study groups, glycemia level, HbA1c, HOMA-IR, insulin level, maternal age, gestational age at sampling and delivery, body weight, gestational weight gain (GWG), body mass index (BMI), inclusion and exclusion criteria for cases and controls, pregnancy maternal and fetal/neonatal outcomes; and (3) newborn characteristics: gender, body weight, body height, neonatal Apgar scores at 1 and 5 min for cases and controls. Data were extracted according to a previously designed protocol by AC. If an eligible manuscript or specific data regarding the lipid profile were unavailable, the authors were contacted to obtain the missing manuscripts or data. The quality assessment and risk of bias of the included observational studies were assessed using the Newcastle-Ottawa tool (NOS) [27]. The Jadad scale was used to measure quality, and ROBINS-I was used to assess the risk of bias for trials. Study quality: According to the NOS criteria, studies were categorized as good quality if they received 3 or 4 stars in the selection domain, 1 or 2 stars in the comparability domain, and 2 or 3 stars in the outcome/exposure domain, or a total score of ≥7 stars. Fair quality was defined as 2 stars in selection, 1 or 2 stars in comparability, and 2 or 3 stars in outcome/exposure, or a total score of 5–6 stars. Poor quality included studies with 0 or 1 star in selection, 0 stars in comparability, or 0 or 1 star in outcome/exposure, or a total score of ≤4 stars. For randomized trials, the Jadad scale was used, with a maximum score of 5 points; studies scoring 0–2 were considered low-quality, and those scoring 3–5 were considered high-quality.

### 2.6. Statistical Analysis

The assessment of the differences in lipid profile parameters (TG, TC, LDL-C, HDL-C, VLDL, and TG/HDL ratio) was defined as the major outcome of this meta-analysis. The overall effect size for the main outcome was assessed using the standardized mean difference (SMD). It was used to measure the differences between the GDM and non-GDM groups. SMD represents the difference between group arithmetic means expressed in units of standard deviation and was calculated by pooling the individual study results using a random-effects model based on the Der Simonian–Laird method. Heterogeneity among studies was evaluated using the Chi-square Q test and I^2^ statistic. The I^2^ value quantifies the proportion of the total variation across studies attributable to true heterogeneity rather than chance. Heterogeneity was categorized according to the Cochrane Handbook [28], with I^2^ values <30%, 30–60%, and >60% indicating low, moderate, and high heterogeneity, respectively. Although forest plots were constructed to illustrate the pooled effect sizes, they were not displayed due to their large dimensions; to summarize the large volume of data, pooled results are presented in summary tables. Complete forest plots for each analysis are available in the Appendix A. Publication bias was assessed using a funnel plot for each defined outcome (Appendix A). Subgroup analysis was performed at each time point according to (1) the region of the respondents’ origin, (2) study design (observational vs. experimental, prospective vs. retrospective vs. cross-sectional), (3) the criteria used for GDM diagnosis, and (4) the presence of smoking and obesity (most commonly reported inclusion/exclusion criteria).

GetData Graph Digitizer ver. 2.26 was used to extract data from figures that presented values of lipid profile parameters in the original article if they were not reported in the manuscript text. Lipid profile parameters were converted into mmol/L as follows: for TC, LDL-C, HDL-C, and VLDL-C given in mg/dL, the equation 1 mmol/L = mg/dL/38.67 was used, and for TG in mg/dL, the equation 1 mmol/L = 1 mg/dL/88.57 was applied. In case of necessity for approximation of arithmetic mean and standard deviation according to other measures of central tendency and measures of variability, we applied the following calculations: (1) median was used as an approximation of arithmetic mean directly, (2) multiple of median (MoM) was converted into arithmetic mean as MoM*median (patient/population value) [29], (3) if IQR was used as a measure of variability, standard deviation (sd) was calculated as sd = IQR/1.35, (4) if standard error (se) was the given variability measure, sd was obtained by the equation sd = se × √*n*, (5) if range was reported, then sd = (max − min)/4, and (6) if 95% confidence interval of the arithmetic mean was known, sd was calculated as √*n* × (Upper limit − Lower limit)/t, where t represents critical value and it is chosen for *n* − 1.

Data were described using absolute and relative numbers. Statistical significance was set at *p* ≤ 0.05. Statistical analyses were conducted using the R programming language and software environment (R4.5.0).

## 3. Results

### 3.1. Systematic Review

Through a comprehensive literature search, 6635 articles were found. After 3284 duplicate articles were removed, 3351 were screened, and 2831 were excluded because they were not in English, were not original articles, had an inappropriate population, had or did not have an inadequate control group, or did not explore TG, TC, LDL-C, HDL-C, VLDL, or TG/HDL ratio. Of the 498 articles reviewed in full text, 457 were eligible for inclusion in the systematic review. The selection process is shown as a flow diagram in Figure 1.

The characteristics of all 457 publications included in the qualitative analysis are presented in detail in Appendix A. They were published between 1980 and 2024, with a total of 188,611 pregnant women, 56,334 with and 132,277 without GDM. The smallest GDM group included five women, and the smallest non-GDM group included 7. The largest GDM group consisted of 2181 women, while the largest non-GDM group consisted of 9067. Of the 457 included studies, the design was reported as retrospective in 124 (27.1%), prospective in 98 (21.4%), and cross-sectional in 50 (10.9%). The study design was not reported in 121 (26.5%) and unclear in 64 (14%). Of all studies with a retrospective design, there were 98/124 case-control, 15/124 nested case-control in prospective cohorts, and 11/124 retrospective cohort studies. There were 94 prospective cohorts and four interventional studies out of 98 with prospective study designs. More than half of the studies originated from Asia (239). The remaining studies were from European countries (159), North America (31), South America (12), Africa (9), and Australia and New Zealand (7). In Asia, the majority of studies originated from China (162 out of 239). In Europe, the highest contributions were from Poland (18 out of 159) and Italy (15 out of 159). In North America, most studies were published in the United States (18 out of 31), while in South America, Brazil accounted for the majority (9 out of 12). In Africa, the highest number of studies came from Egypt (4 out of 9).

A total of 34 different criteria were used for GDM diagnosis in the included original articles. Most of these criteria use some form of OGTT for a definitive diagnosis. The most commonly applied criterion was the IADPSG, which was used in 154 of 457 studies. Other diagnostic guidelines included the ADA (89 studies), WHO (32), Carpenter and Coustan (27), NDDG (25), O’Sullivan (5), Chinese Medical Association (5), ADIPS (4), Chinese Guideline (4), Ministry of Health of China (3), DIPSI (2), Austrian Diabetes Association (2), Polish Diabetes Association (2), ACOG (1), HAPO (1), and others (19). Notably, 20% of the studies did not report the diagnostic criteria used for GDM. All details regarding the criteria and definitions used for GDM diagnosis are provided in Appendix A.

The characteristics of the GDM cases and non-GDM controls in the included studies are shown in Appendix A. The status of glucose metabolism parameters (glycemia, HbA1c, insulin, and HOMA-IR) in groups of pregnant women with and without GDM was not reported in a large number of studies. Blood glucose levels were the most frequently reported parameter (85% in the GDM group vs. 85% in the non-GDM group), followed by HbA1c (47% vs. 44%), insulin (39% vs. 38%), and HOMA-IR, which was the least commonly reported (38% vs. 38%). The important characteristics of the cases and controls that could influence the effect of lipid profile on GDM appearance, maternal age, body weight, gestational weight gain (GWG), and body mass index (BMI) were extracted from the included studies. Maternal age and BMI were reported in the majority of studies (94% and 91% in the GDM and non-GDM groups), whereas body weight and gestational weight gain (GWG) were reported in only 18% of the included studies in both groups. Study groups were matched in 337/457 studies, unmatched in 45/457 studies, and matching was not reported in 75/457 studies. Maternal age and pre-pregnancy BMI were the most commonly used parameters for matching (265/457 and 224/457, respectively). The inclusion and exclusion criteria are presented in Appendix A. A large proportion of studies considered only singleton pregnancies (84.5%). Pregnant women with previous cardiovascular, renal, and liver diseases were not included in any of the studies, while women with a family history of DM, obese women, women with a family history of GDM, nulliparous women, women with hypertension, and women with pre-existing DM were included in 18%, 12%, 3.5%, 3%, 1% and 1% of the studies, respectively.

Adverse pregnancy outcomes and perinatal maternal and neonatal complications were rarely reported (Appendix A). The gestational age at delivery was not reported in 78% of the cases in both study groups. Adverse pregnancy outcomes were reported as follows: macrosomia in 47 and 44 cases, preterm delivery in 21 and 21, small for gestational age (SGA) in 21 and 19, preeclampsia (PE) in 19 and 14, pregnancy-induced hypertension (PIH) in 14 and 11, neonatal hypoglycemia in 4 and 4, hyperbilirubinemia in 2 and 2, respiratory distress syndrome in 3 and 2, birth trauma in 2 and 2, stillbirth in 2 and 0, elevated liver enzymes in 1 and 1, fetal or neonatal loss in 2 and 1, and congenital anomalies in 1 and 1 cases, respectively, across a total of 457 studies. Neonatal characteristics are presented in (Appendix A). Data regarding neonatal sex were collected in only four of the 457 studies. There were 166 male and 176 female newborns in the GDM group and 691 male and 552 female newborns in the non-GDM group. The minimum average newborn body weight from GDM pregnancies was 1850 g, and from non-GDM pregnancies was 2750 g, while the maximum was 4750 g and 3892.6 g, respectively.

### 3.2. Quality Assessment

Most of the included studies that clearly reported their design were of good quality (209/272). There were 45 studies with fair quality and 18 with poor quality (Appendix A).

### 3.3. Meta-Analysis

This meta-analysis was performed for lipid profile parameters: TG, TC, LDL-C, HDL-C, VLDL, and TG/HDL ratio measurements taken during the 1st, 2nd, and 3rd trimesters according to the available data from the included publications (Appendix A). Subsequently, a meta-regression analysis of maternal age and BMI was conducted. Finally, a subgroup analysis was carried out at each time point according to: (1) the region of the respondents’ origin, (2) study design (observational vs. experimental, prospective vs. retrospective vs. cross-sectional), (3) the criteria used for GDM diagnosis, and (4) the presence of smoking and obesity (most commonly reported inclusion/exclusion criteria).

The lipid profile parameters in GDM and non-GDM pregnancies are shown in Table 2. The meta-analysis demonstrated distinct trimester-specific alterations in maternal lipid profiles during pregnancy. TG levels were significantly elevated across all three trimesters, with a gradual decrease in effect size as pregnancy progressed (SMD = 0.61, SMD = 0.57, and SMD = 0.48 in the 1st, 2nd, and 3rd trimesters, respectively; *p* < 0.001 for all) (Appendix A). Similarly, the TG/HDL ratio was significantly increased throughout pregnancy, showing the strongest elevation in the second trimester (SMD = 0.44, SMD = 0.66, and SMD = 0.49 for the 1st, 2nd, and 3rd trimester, respectively; *p* < 0.001 for all) (Appendix A). TC and LDL-C levels were significantly increased during the first and second trimesters (*p* < 0.001 in the 1st and 2nd trimesters for both parameters) (Appendix A). However, these changes were no longer statistically significant in the third trimester (Appendix A). HDL-C concentrations were significantly reduced in all three trimesters, with relatively stable effect sizes (SMD = −0.32, −0.32, and −0.30, respectively; *p* < 0.001) (Appendix A). VLDL levels, on the other hand, showed a statistically significant increase only in the second trimester (SMD = 0.89, *p* < 0.001) (Appendix A), whereas the changes observed in the first and third trimesters were not significant (SMD = 0.67, *p* = 0.064; SMD = 0.28, *p* = 0.154, respectively) (Appendix A).

### 3.4. Meta-Regression

To assess the potential influence of maternal age and BMI on lipid profile alterations in GDM, meta-regression analysis was performed across all three trimesters (Table 3). In the first trimester, maternal age was significantly associated with TC levels (β = 0.250; 95% CI: 0.03 to 0.47; *p* = 0.023) and BMI was positively associated with TC and LDL (β = 0.332; 95% CI: 0.20 to 0.46; *p* < 0.001 and β = 0.116; 95% CI: 0.02 to 0.22; *p* = 0.023, respectively). In the second trimester, maternal age had a statistically significant but small positive association with TG levels (β = 0.021; 95% CI: –0.02 to 0.06; *p* < 0.001) and a significant negative association with HDL (β = –0.027; 95% CI: –0.08 to –0.03; *p* < 0.001), indicating that older age was linked to lower HDL-C levels. Notably, the TG/HDL-C ratio also increased with age (β = 0.182; 95% CI: 0.01 to 0.35; *p* = 0.034). BMI was significantly positively associated with TG (β = 0.066; 95% CI: 0.03 to 0.10; *p* < 0.001) and VLDL (β = 0.548; 95% CI: 0.14 to 0.95; *p* = 0.008). During the third trimester, no significant associations were observed between age or BMI and lipid parameters.

### 3.5. Subgroup Analysis

First, a subgroup meta-analysis was performed to evaluate the regional differences in lipid profile parameters among pregnant women with and without GDM (Table 4). The studies were classified into five geographical regions: Asia, Europe, America (North and South America), Africa, and Australia and New Zealand. Across all trimesters, studies from Asia consistently demonstrated elevated TG, TC, and LDL-C levels and decreased HDL-C levels in GDM compared to non-GDM pregnancies. In European cohorts, TG levels were also significantly increased, and HDL-C levels decreased in all trimesters. TC and LDL-C elevations were modest and even reversed in the third trimester. Studies from America have shown inconsistent trends in lipid profile levels throughout pregnancy. Due to the scarcity of studies from Africa, Australia, and New Zealand, the results were irrelevant.

There was a significant difference in the assessed overall effect size between geographical regions for TC and LDL-C (*p* = 0.011 and *p* = 0.028, respectively) during the 1st trimester, for TC, LDL-C, VLDL, and TG/HDL ration (*p* = 0.012, *p* < 0.001, *p* = 0.008 and *p* = 0.006, respectively) during the 2nd, and for TG, TC, LDL-C, and VLDL (*p* = 0.005, *p* < 0.001, *p* = 0.032 and *p* = 0.031, respectively) during the 3rd trimester.

Second, we performed a subgroup meta-analysis to evaluate the differences in lipid profile parameters between GDM and non-GDM pregnancies across different study designs: retrospective, prospective, and cross-sectional designs (Table 5). Retrospective studies have shown a consistent trend of dyslipidemia in women with GDM compared to non-GDM pregnancies, with elevated TG, TC, and LDL-C and decreased HDL-C values across all trimesters. In prospective studies, elevated TG levels were observed in all trimesters. Unlike retrospective studies, TC and LDL-C increases were more notable in the first trimester but diminished or reversed in later pregnancies. HDL-C levels were consistently lower in GDM pregnancies. Cross-sectional studies presented much more variability.

There was a significant difference in the assessed overall effect size between different study designs for TG and TC (*p* = 0.030 and *p* = 0.023, respectively) during the 1st trimester, for HDL-C and VLDL and TG/HDL ratio (*p* = 0.004, *p* < 0.001, and *p* = 0.001, respectively) during the 2nd trimester, and for TC and LDL-C (*p* = 0.004 and *p* = 0.014, respectively) during the 3rd trimester.

Third, the subgroup meta-analysis of lipid profile alterations in GDM and non-GDM pregnancies categorized by diagnostic criteria used in the original articles (IADPSG, ADA, WHO) per trimester revealed consistent trends across studies (Table 6). TG levels were significantly elevated across all trimesters and diagnostic criteria. TC levels were also elevated across all trimesters if GDM was diagnosed according to the IADPSG and ADA diagnostic criteria, as well as in the 1st and 3rd trimesters when diagnosed according to the WHO criteria. LDL-C levels were significantly elevated in the 1st and 2nd trimesters when the IADPSG and ADA criteria were applied. HDL-C was reduced in the 1st when the IADPSG and WHO criteria were used, and in the 2nd, regardless of the GDM criteria used. VLDL and TG/HDL ratio data were limited due to the small number of included studies. There was a significant difference in the assessed overall effect size according to the applied diagnostic criteria for TG, TC, and LDL-C (*p* = 0.037, *p* = 0.004, and *p* = 0.039, respectively) during the 1st trimester. There were no differences in any of the evaluated lipid parameters according to the diagnostic criteria during the 2nd and 3rd trimesters.

Finally, a subgroup meta-analysis was conducted to explore the impact of smoking and obesity on lipid profile differences in the 1st, 2nd, and 3rd trimesters of pregnancy (Table 7). Four subgroups were analyzed: (1) studies including both smokers and non-smokers, (2) studies including only non-smokers, (3) studies including both obese and non-obese pregnant women, and (4) studies including obese pregnancies exclusively. The results showed that: (1) TG levels were significantly higher in pregnant women with GDM across pregnancy, regardless of whether they were exclusively non-smokers or included both smokers and non-smokers, and regardless of whether the study population included only non-obese or both non-obese and obese pregnancies; (2) TC levels were higher in pregnant women in the 1st and 2nd trimester (except in non-smokers exclusively) regardless of smoking or nutritional status; (3) LDL-C was significantly higher, and HDL-C significantly lower in the first and second trimesters among pregnant women with GDM, regardless of whether they were smokers, non-smokers, or non-obese. However, when obese pregnant women were included, LDL-C levels did not differ significantly between GDM and non-GDM pregnancies, while HDL-C levels were significantly lower in all three trimesters. VLDL and TG/HDL ratio data were limited due to the small number of studies.

## 4. Discussion

This review systematically examines how GDM influences lipid profile variations during pregnancy, with an emphasis on trimester-specific patterns, specifically focusing on parameters such as TG, TC, LDL-C, HDL-C, VLDL, and TG/HDL ratios. Our findings underscore significant alterations in lipid metabolism during pregnancy in women with GDM compared to non-GDM pregnancies. TG and TG/HDL ratio were significantly higher across all trimesters, TC and LDL-C in the 1st and 2nd, and VLDL in the 2nd only, while HDL-C was significantly lowered during the whole pregnancy in GDM pregnant women. In pregnancies that will develop GDM, TC and LDL-C levels increase in the 1st and 2nd trimesters, which reflects early insulin resistance and lipid metabolism disorders. Insulin resistance stimulates the liver to increase lipid synthesis through the activation of the enzyme HMG-CoA reductase and transcription factor SREBP-1c. These changes, additionally stimulated by placental hormones, lead to an increase in TC and LDL-C. This change in the lipid profile can potentially serve as an early risk indicator for GDM and as the basis for implementing preventive measures before the onset of clinical hyperglycemia [30,31].

Physiological pregnancy induces insulin resistance and hyperlipidemia as adaptive responses to fetal growth demands; these changes are markedly accentuated in GDM [32]. Hypertriglyceridemia begins in early pregnancy, intensifies through the 2nd and 3rd trimesters, and remains significantly higher in late gestation and term in women with GDM [33,34]. Our analysis demonstrated that in women with GDM, TG levels were consistently elevated across all three trimesters, with the highest elevation observed during the first trimester. Elevated TG levels are expected in physiological pregnancy, as a result of the hormonal and metabolic adaptation, but our result proved that levels of TGs were significantly higher in GDM compared to non-GDM pregnancies. The mechanism starts with enhanced insulin resistance, which is the basis of GDM, combined with increased hepatic triglyceride production as a result of higher glycemia levels and insulin resistance, which leads to de novo liver lipogenesis (VLDL) and impaired lipid clearance mechanisms [35]. Additionally, human placental lactogen (HPL), placental growth hormone, and progesterone reduce maternal adipose lipoprotein lipase (LPL) activity while increasing placental LPL expression. This enhances TG hydrolysis at the fetal-placental membrane, directing free fatty acids (FFA) towards the fetus. These effects are increased in patients with GDM [36,37]. In a prospective cohort study, Zhu et al. reported that pregnant women who later developed GDM had significantly higher TG levels as early as weeks 6–8. of pregnancy (1.2 vs. 1.0 mmol/L) and a greater increase in TG until 16–18 weeks of pregnancy (0.8 vs. 0.7 mmol/L) compared to healthy pregnant women. High early TG levels and their accelerated increase were associated with a two-fold higher risk of developing GDM, independent of age, BMI, and glycemia [38]. In studies that explored the same variables but in mid-gestation and late 3rd trimester the results were consistent; Wang et al. showed that in the second trimester, pregnant women with GDM had significantly higher TG values than healthy pregnant women (2.57 ± 1.13 vs. 2.23 ± 0.93 mmol/L; *p* < 0.001). This difference persisted in the third trimester, where TG values were 3.36 ± 1.51 mmol/L in the GDM group compared to 3.08 ± 1.23 mmol/L in the control group (*p* < 0.001) [39]. Similarly, results of Bharathi KR et al. (3.23 ± 0.88 vs. 1.88 ± 0.29 mmol/L, *p* < 0.001) validate the association between hypertriglyceridemia in mid-pregnancy and the development of GDM [40].

Similarly, the TG/HDL ratio was significantly higher throughout pregnancy, with the strongest elevation observed during the second trimester. The TG/HDL ratio is a key indicator of lipid metabolism and is commonly used as a marker of insulin resistance and metabolic disturbances [41]. This ratio reflects the balance between TG and HDL-C, thus providing insight into the overall lipid profile and metabolic health of an individual, with higher ratios suggesting a greater risk of metabolic disturbances and cardiovascular events [42]. In GDM, the TG/HDL ratio is often elevated due to the physiological and metabolic changes that occur during pregnancy [43]. These changes include increased insulin resistance and alterations in lipid metabolism. Studies have shown that the TG/HDL-C ratio is a potential biomarker for identifying women at risk of GDM early in pregnancy. Elevated TG/HDL ratios in early pregnancy are associated with an increased risk of developing GDM [44]. In obesity, the TG/HDL ratio is also elevated. Obesity is characterized by chronic low-grade inflammation, insulin resistance, and dyslipidemia, all of which contribute to an unfavorable lipid profile. The TG/HDL-C ratio has been used to assess the risk of metabolic syndrome, type 2 diabetes, and cardiovascular diseases, all of which are prevalent in obese individuals [45]. The ratio serves as a simple yet effective indicator of metabolic dysfunction, which is why it is commonly used in clinical practice to monitor and predict the onset of GDM, metabolic syndrome, and cardiovascular disease. Our findings of an elevated TG/HDL-C ratio are consistent with the pathophysiological changes that characterize insulin resistance, a hallmark of GDM, which leads to lipid dysregulation. Increased triglyceride levels and imbalances in lipid ratios may intensify insulin resistance associated with GDM, potentially increasing the likelihood of cardiovascular complications later in life for both mothers and their offspring [35,46].

The observed elevation in TG levels and consistent reduction in HDL-C levels in GDM pregnancies are consistent with previous meta-analyses and cohort studies [44,47,48]. Shi et al., (2023) [49] reported that TC, TG, LDL-C, and HDL-C increased significantly from the second to the third trimester in both GDM and non-GDM groups (*p* < 0.001). Compared with women without GDM, those with GDM exhibited consistently higher TC and TG levels in both trimesters. Specifically, the median TG concentration in the GDM group increased from 2.52 mmol/L (interquartile range [IQR]: 1.79–2.92) in the second trimester to 3.83 mmol/L (IQR: 2.32–4.58) in the third trimester. In contrast, HDL levels were significantly lower among women with GDM, declining from 1.79 mmol/L (IQR: 1.51–2.25) in the second trimester to 1.62 mmol/L (IQR: 1.41–2.01) in the third trimester. No significant differences were observed in LDL-C levels between the GDM and non-GDM groups in either trimester [49]. Furthermore, our study supports the findings of Chodick et al., who reported significantly higher lipid profiles, which could be a key factor associated with macro- and micro-vascular complications in women with GDM [50]. The reduction in HDL-C levels observed across all trimesters aligns with studies linking low HDL-C levels to increased inflammation and vascular dysfunction, which are prevalent in women with GDM. HDL-C is traditionally recognized for its atheroprotective and anti-inflammatory properties, primarily due to its role in reverse cholesterol transport, antioxidative activity, and inhibition of vascular adhesion molecules. In the context of GDM, decreased HDL-C levels compromise these protective functions, contributing to a pro-inflammatory milieu [46]. The diminished HDL-C levels observed in GDM are closely associated with elevated levels of pro-inflammatory cytokines (e.g., TNF-α and IL-6) and acute-phase reactants, such as C-reactive protein (CRP). These inflammatory mediators not only downregulate apolipoprotein A-I synthesis, an essential structural component of HDL-C, but also promote HDL-C oxidation and impair its function. Concurrent hypertriglyceridemia, another hallmark of GDM, further exacerbates HDL-C depletion through the cholesteryl ester transfer protein (CETP)-mediated exchange of lipids, leading to the formation of small, dense HDL-C particles with reduced anti-atherogenic capacity [51]. This interplay between dyslipidemia and inflammation fosters a metabolic environment conducive to insulin resistance, which is both a cause and a consequence of GDM. Thus, a low HDL-C level is both a marker and mediator of dysregulated metabolism [52].

However, our study also highlighted some regional differences in lipid alterations in GDM pregnancies. Studies from Asia and Europe showed consistently elevated TG and TC levels, while studies from North America exhibited more variable trends. These regional differences may be attributed to variations in dietary habits, genetic factors, and diagnostic criteria, which have been shown to influence lipid metabolism in different populations. Asian diets, which are often high in carbohydrates and refined grains, may promote hepatic de novo lipogenesis and TG accumulation [53]. Western diets, which are rich in saturated fats and low in fiber, can lead to broader dyslipidemia but may affect different lipid subtypes (e.g., apoB and sdLDL). Mediterranean diets, which are high in unsaturated fats and antioxidants, are associated with protective lipid profiles and lower GDM risk [15]. For example, South Asians are genetically predisposed to higher TG and lower HDL-C levels, independent of body weight [54]. This variability underscores the need for region-specific clinical guidelines for managing lipid levels in GDM.

Our study showed that maternal age and pre-pregnancy BMI are associated with significant changes in lipid status, but not necessarily for all evaluated lipid parameters, and not with the same pattern across all three trimesters. Women with higher pre-pregnancy BMI tended to exhibit more pronounced atherogenic lipoprotein levels, especially elevated TG and reduced HDL-C levels, across all three trimesters. This dyslipidemia pattern, exacerbated by insulin resistance, increases the risk of preeclampsia, macrosomia, and other metabolic complications [55]. Advanced maternal age is also associated with a decline in insulin sensitivity, which further amplifies lipid metabolism abnormalities, contributing to higher maternal and fetal morbidity [56]. Patients with GDM and obesity face a higher lifetime risk of cardiovascular disease (CVD), primarily due to persistent insulin resistance and dyslipidemia after pregnancy. Elevated TG levels, low HDL-C levels, and increased LDL-C levels contribute to atherosclerosis, increasing the risk of coronary artery disease, stroke, and hypertension in later life [57]. Offspring of mothers with GDM or obesity are at an increased risk of developing metabolic disorders, such as obesity, insulin resistance, and T2DM, later in life. This is likely due to epigenetic changes and metabolic alterations induced by the maternal metabolic environment during pregnancy, which affect fetal development and predispose offspring to these conditions [58]. The role of lipid-modifying interventions, such as dietary changes and pharmacological treatments, remains of significant interest. Given the persistence of dyslipidemia in GDM pregnancies, future research should explore the efficacy of statins, omega-3 fatty acids, and lifestyle modifications in improving lipid profiles and reducing adverse pregnancy outcomes [59,60].

### Strengths and Limitations

A notable strength of this review is its extensive dataset, which encompasses lipid profile information from more than 188,000 pregnant individuals, allowing for robust statistical analysis. The meta-regression performed to assess the influence of maternal age and BMI on lipid abnormalities in GDM is another strength, providing valuable insights into how these maternal characteristics contribute to lipid dysregulation. Additionally, our study utilized a rigorous and systematic search strategy, ensuring that relevant studies were included and potential biases were minimized. Despite the aforementioned strengths, this study has several limitations. The most significant methodological limitation is the heterogeneity observed across studies, particularly in the timing of lipid measurements, diagnostic criteria for GDM, and study design (cross-sectional, prospective, retrospective). This variability makes it difficult to generalize the findings to the entire population. An important limitation of this analysis is the extensive number of subgroup analyses, which raises concerns about inflated type I error rates, increasing the likelihood of false-positive findings; therefore, these results warrant cautious interpretation and independent replication. Additionally, many studies did not report important confounding factors, such as pre-pregnancy lipid levels, gestational weight gain, or pre-existing medical conditions, which could have influenced lipid changes during pregnancy. The lack of consistent reporting on perinatal outcomes further limits our ability to fully evaluate the clinical significance of lipid alterations in GDM pregnancies. Furthermore, the regional differences in study designs and patient populations, particularly from regions like Africa and Oceania, where limited data are available, point to the need for further research in these regions. Future studies should aim to standardize methodologies, including lipid measurement techniques and diagnostic criteria, to enhance the consistency and applicability of the findings across different settings. Future investigations should prioritize longitudinal cohort designs that monitor lipid levels from the preconception phase through each trimester to better understand the temporal metabolic changes. Additionally, more studies are needed to examine the potential impact of lipid-modifying interventions on maternal and fetal outcomes in pregnancies with GDM. Research exploring the genetic basis of lipid dysregulation in GDM could also help identify individuals at the highest risk and lead to more personalized treatment options. Finally, further exploration of the relationship between lipid profiles and long-term cardiovascular health in women with GDM is needed.

## 5. Conclusions

In conclusion, this systematic review and meta-analysis provide strong evidence that GDM is associated with significant lipid abnormalities, and TG and the TG/HDL-C ratio were consistently elevated in GDM pregnancies across all trimesters, with the strongest differences observed in the first trimester (SMD 0.61 and 0.44), highlighting their potential as early metabolic indicators. TC and LDL-C levels were significantly higher only in the first and second trimesters, then plateaued, indicating that their diagnostic relevance is largely limited to early and mid-gestation. HDL-C levels were significantly lower throughout pregnancy, serving as a stable inverse marker of dyslipidemia. VLDL-C levels showed a distinct peak in the second trimester (SMD 0.89), corresponding to the period of peak insulin resistance. Given the impact of lipid profiles on maternal and fetal outcomes, regular lipid screening in high-risk pregnancies should be considered part of comprehensive care for women with or at risk of GDM. The overall results emphasized the importance of early lipid monitoring in all pregnant women, with particular attention to the identified at-risk groups and older pregnancies with a higher BMI. Finally, a multidisciplinary medical approach is essential for maintaining and safeguarding maternal and fetal health in pregnancies complicated by dyslipidemia.

## Figures and Tables

**Figure 1 medicina-61-01290-f001:**
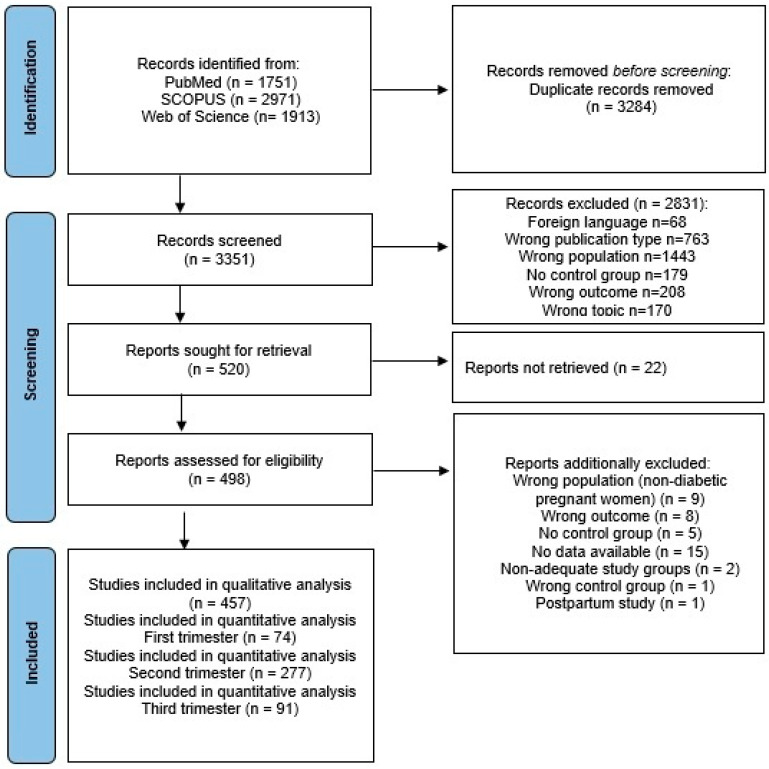
PRISMA flow diagram.

**Table 1 medicina-61-01290-t001:** Search strategies.

Database	Search Query
**PubMed**	(“Diabetes, Gestational” or “Diabetes Mellitus, Gestational” or “Diabetes, Pregnancy-Induced” or “Diabetes, Pregnancy induced” or “Gestational Diabetes” or “GDM”) and (“Triglyceride*” or “Triacylglycerol*” or “TGs” or “Tgs” or “Total cholesterol” or “TC” or “Tc” or “Heavy Lipoproteins” or “High Density Lipoprotein*” or “High-Density Lipoprotein*” or “alpha-1 Lipoprotein” or “alpha-Lipoprotein*” or “HDL” or “HDL-C” or “LDL(1)” or “LDL(2)” or “LDL-1” or “LDL-2” or “LDL1” or “LDL2” or “Low-Density Lipoprotein*” or “Low-Density Lipoprotein 1” or “Low-Density Lipoprotein 2” or “beta-Lipoprotein*” or “LDL” or “LDL-C” or “cholesterol” or “epicholesterol” or “Dyslipid*” or “Hyperlipid*” or “Hypertriglycerid*” or “Hypercholesterol*” or “Lipid status” or “Lipid profile” or “Lipid blood test*” or “Lipid panel*” or “Lipid disorder*”)
**Web of Science**	(TITLE-ABS-KEY (“Diabetes, Gestational”) OR TITLE-ABS-KEY (“Diabetes Mellitus, Gestational”) OR TITLE-ABS-KEY (“Diabetes, Pregnancy-Induced”) OR TITLE-ABS-KEY (“Diabetes, Pregnancy induced”) OR TITLE-ABS-KEY (“Gestational Diabetes”) OR TITLE-ABS-KEY (“GDM”)) AND (TITLE-ABS-KEY (“Triglyceride*”) OR TITLE-ABS-KEY (“Triacylglycerol*”) OR TITLE-ABS-KEY (“TGs”) OR TITLE-ABS-KEY (“Tgs”) OR TITLE-ABS-KEY (“Total cholesterol”) OR TITLE-ABS-KEY (“TC”) OR TITLE-ABS-KEY (“Tc”) OR TITLE-ABS-KEY (“Heavy Lipoproteins”) OR TITLE-ABS-KEY (“High Density Lipoprotein*”) OR TITLE-ABS-KEY (“High-Density Lipoprotein*”) OR TITLE-ABS-KEY (“alpha-1 Lipoprotein”) OR TITLE-ABS-KEY (“alpha-Lipoprotein*”) OR TITLE-ABS-KEY (“HDL”) OR TITLE-ABS-KEY (“HDL-C”) OR TITLE-ABS-KEY (“LDL(1)”) OR TITLE-ABS-KEY (“LDL(2)”) OR TITLE-ABS-KEY (“LDL-1”) OR TITLE-ABS-KEY (“LDL-2”) OR TITLE-ABS-KEY (“LDL1”) OR TITLE-ABS-KEY (“LDL2”) OR TITLE-ABS-KEY(“Low-Density Lipoprotein*”) OR TITLE-ABS-KEY (“Low-Density Lipoprotein 1”) OR TITLE-ABS-KEY (“Low-Density Lipoprotein 2”) OR TITLE-ABS-KEY (“beta-Lipoprotein*”) OR TITLE-ABS-KEY (“LDL”) OR TITLE-ABS-KEY (“LDL-C”) OR TITLE-ABS-KEY (“cholesterol”) OR TITLE-ABS-KEY (“epicholestero”) OR TITLE-ABS-KEY (“Dyslipid*”) OR TITLE-ABS-KEY (“Hyperlipid*”) OR TITLE-ABS-KEY (“Hypertriglycerid*”) OR TITLE-ABS-KEY (“Hypercholesterol*”) OR TITLE-ABS-KEY (“Lipid status”) OR TITLE-ABS-KEY (“Lipid profile”) OR TITLE-ABS-KEY (“Lipid blood test*”) OR TITLE-ABS-KEY (“Lipid panel*”) OR TITLE-ABS-KEY (“Lipid disorder*”))
**SCOPUS**	(ALL=(“Diabetes, Gestational”) OR ALL=(“Diabetes Mellitus, Gestational”) OR ALL=(“Diabetes, Pregnancy-Induced”) OR ALL=(“Diabetes, Pregnancy induced”) OR ALL=(“Gestational Diabetes”) OR ALL=(“GDM”)) AND (ALL=(“Triglyceride*”) OR ALL=(“Triacylglycerol*”) OR ALL=(“TGs”) OR ALL=(“Tgs”) OR ALL=(“Total cholesterol”) OR ALL=(“TC”) OR ALL=(“Tc”) OR ALL=(“Heavy Lipoproteins”) OR ALL=(“High Density Lipoprotein*”) OR ALL=(“High-Density Lipoprotein*”) OR ALL=(“alpha-1 Lipoprotein”) OR ALL=(“alpha-Lipoprotein*”) OR ALL=(“HDL”) OR ALL=(“HDL-C”) OR ALL=(“LDL(1)”) OR ALL=(“LDL(2)”) OR ALL=(“LDL-1”) OR ALL=(“LDL-2”) OR ALL=(“LDL1”) OR ALL=(“LDL2”) OR ALL=(“Low-Density Lipoprotein*”) OR ALL=(“Low-Density Lipoprotein 1”) OR ALL=(“Low-Density Lipoprotein 2”) OR ALL=(“beta-Lipoprotein*”) OR ALL=(“LDL”) OR ALL=(“LDL-C”) OR ALL=(“cholesterol”) OR ALL=(“epicholestero*”) OR ALL=(“Dyslipid*”) OR ALL=(“Hyperlipid*”) OR ALL=(“Hypertriglycerid*”) OR ALL=(“Hypercholesterol*”) OR ALL=(“Lipid status”) OR ALL=(“Lipid profile”) OR ALL=(“Lipid blood test*”) OR ALL=(“Lipid panel*”) OR ALL=(“Lipid disorder*”))

* asterisk is a wildcard character used in search strategies to represent one or more characters in order to broaden search results.

**Table 2 medicina-61-01290-t002:** Meta-analysis of lipid profile parameters (TG, TC, LDL, HDL, VLDL, and TG/HDL ratio) in pregnant women with and without GDM according to the time of measurement.

Lipid Profile Parameter	I^2^ (%)	1st Trimester			I^2^ (%)	2nd Trimester			I^2^ (%)	3rd Trimester		
		SMD	95%CI SMD	*p* *		SMD	95%CI SMD	*p*		SMD	95%CI SMD	*p*
**TG**	95.3	0.61	0.52–0.69	**<0.001**	91.9	0.57	0.51–0.63	**<0.001**	88.7	0.48	0.39–0.57	**<0.001**
**TC**	94.6	0.38	0.30–0.47	**<0.001**	93.5	0.27	0.20–0.33	**<0.001**	91.3	0.04	−0.06–0.14	0.409
**LDL-C**	94.1	0.32	0.24–0.41	**<0.001**	91.3	0.20	0.14–0.26	**<0.001**	93.1	0.01	−0.10–0.12	0.862
**HDL-C**	95.6	−0.32	−0.41 to −0.22	**<0.001**	91.6	−0.31	−0.38 to −0.25	**<0.001**	96.8	−0.30	−0.46 to −0.13	**<0.001**
**VLDL-C**	90.6	0.67	−0.04–1.38	0.064	93.3	0.89	0.53–1.26	**<0.001**	87.6	0.28	−0.10–0.66	0.154
**TG/HDL ratio**	77.3	0.44	0.34–0.54	**<0.001**	87.9	0.66	0.41–0.91	**<0.001**	0.0	0.49	0.37–0.61	**<0.001**

* For a significance level of 0.05, according to the random effects model meta-analysis with Standardized Mean Difference (SMD) as the overall effect size measurement. *p*-values that reached statistical significance are shown in bold.

**Table 3 medicina-61-01290-t003:** Meta-regression of maternal age and BMI on lipid profile parameters in the 1st, 2nd, and 3rd trimesters.

Trimester	Lipid Profile Parameter	Δ Age			Δ BMI		
		β	95%CI β	*p*	β	95%CI β	*p*
**1st**	TG	−0.025	−0.19–0.14	0.766	0.066	−0.03–0.17	0.192
	TC	0.250	0.03–0.47	**0.023**	0.332	0.20–0.46	**<0.001**
	LDL-C	0.129	−0.04–0.30	0.138	0.116	0.02–0.22	**0.023**
	HDL-C	0.217	−0.06–0.49	0.119	−0.032	−0.20–0.13	0.705
	VLDL-C	−0.369	−0.93–0.19	0.197	−0.095	−0.24–0.05	0.197
	TG/HDL ratio						
**Trimester**	**Lipid profile parameter**	**Δ Age**			**Δ BMI**		
		**β**	**95%CI β**	** *p* **	**β**	**95%CI β**	** *p* **
**2nd**	TG	0.021	−0.02–0.06	**<0.001**	0.066	0.03–0.10	**<0.001**
	TC	−0.023	−0.08–0.04	0.438	0.040	−0.01–0.09	0.122
	LDL-C	−0.043	−0.10–0.01	0.140	0.010	−0.04–0.06	0.673
	HDL-C	−0.027	−0.08–0.03	**<0.001**	0.013	−0.03–0.06	**<0.001**
	VLDL-C	−0.601	−1.7–0.50	0.286	0.548	0.14–0.95	**0.008**
	TG/HDL ratio	0.182	0.01–0.35	**0.034**	0.058	−0.22–0.33	0.681
**Trimester**	**Lipid profile parameter**	**Δ Age**			**Δ BMI**		
		**β**	**95%CI β**	** *p* **	**β**	**95%CI β**	** *p* **
**3rd**	TG	−0.01	−0.11–0.08	**<0.001**	−0.02	−0.10–0.07	**<0.001**
	TC	0.01	−0.09–0.10	0.938	−0.03	−0.11–0.06	0.570
	LDL-C	0.01	−0.14–0.15	0.991	−0.02	−0.14–0.09	0.711
	HDL-C	0.09	−0.03–0.22	0.154	0.02	−0.08–0.13	0.696
	VLDL-C	−0.11	−0.30–0.09	0.288	−0.106	−0.30–0.09	0.288
	TG/HDL ratio	−0.05	−0.34–0.24	**<0.001**	−0.05	−0.34–0.24	**<0.001**

*p*-values that reached statistical significance are shown in bold.

**Table 4 medicina-61-01290-t004:** Meta-analysis of lipid profile parameters (TG, TC, LDL, HDL, VLDL, and TG/HDL ratio) in GDM compared to non-GDM pregnancies according to geographical region.

Lipid Profile Parameter	Asia								
	1st Trimester			2nd Trimester			3rd Trimester		
	SMD	95%CI SMD	k	SMD	95%CI SMD	k	SMD	95%CI SMD	k
TG	**0.61**	**0.52–0.71**	63	**0.58**	**0.50–0.66**	138	**0.68**	**0.53–0.82**	49
TC	**0.34**	**0.25–0.43**	57	**0.28**	**0.20–0.37**	130	**0.28**	**0.11–0.44**	47
LDL-C	**0.31**	**0.21–0.41**	55	**0.25**	**0.17–0.32**	120	**0.20**	**0.02–0.39**	45
HDL-C	**−0.28**	**−0.39 to −0.17**	56	**−0.34**	**−0.42 to −0.26**	121	**−0.35**	**−0.61 to −0.10**	45
VLDL-C	NA	NA	2	**1.41**	**0.69–2.13**	9	0.39	−0.10–0.89	7
TG/HDL ratio	**0.44**	**0.34–0.54**	8	**0.80**	**0.01–0.56**	9	NA	NA	1
	**Europe**								
**Lipid profile parameter**	**1st trimester**			**2nd trimester**			**3rd trimester**		
	**SMD**	**95%CI SMD**	**k**	**SMD**	**95%CI SMD**	**k**	**SMD**	**95%CI SMD**	**k**
TG	**0.66**	**0.45–0.86**	14	**0.48**	**0.38–0.58**	100	**0.33**	**0.21–0.45**	54
TC	**0.34**	**0.24–0.43**	13	**0.20**	**0.09–0.32**	95	−0.06	−0.16–0.04	49
LDL-C	**0.34**	**0.22–0.46**	10	**0.14**	**0.05–0.24**	88	**−0.14**	**−0.25 to −0.02**	41
HDL-C	**−0.21**	**−0.34 to −0.09**	11	**−0.27**	**−0.38 to −0.16**	91	**−0.10**	**−0.24–0.04**	44
VLDL-C	NA	NA	1	**0.69**	**0.22–1.15**	9	NA	NA	1
TG/HDL ratio	NA	NA	0	NA	NA	1	NA	NA	0
	**North and South America**								
**Lipid profile parameter**	**1st trimester**			**2nd trimester**			**3rd trimester**		
	SMD	95%CI SMD	k	SMD	95%CI SMD	k	SMD	95%CI SMD	k
TG	0.39	−0.07–0.84	9	**0.77**	**0.54–0.99**	25	**0.43**	**0.16–0.70**	13
TC	**0.15**	**0.07–0.24**	9	0.22	−0.07–0.51	23	−0.12	−0.17–0.03	13
LDL-C	0.01	−0.21–0.23	8	**−0.21**	**−0.40 to −0.02**	19	−0.12	−0.29–0.06	11
HDL-C	**−0.77**	**−1.45 to −0.08**	8	−0.21	−0.43–0.01	19	**−0.81**	**−1.47 to −0.14**	11
VLDL-C	NA	NA	1	0.26	−0.02–0.54	3	NA	NA	2
TG/HDL ratio	NA	NA	0	NA	NA	2	NA	NA	2
	**Africa**								
**Lipid profile parameter**	**1st trimester**			**2nd trimester**			**3rd trimester**		
	**SMD**	**95%CI SMD**	**k**	**SMD**	**95%CI SMD**	**k**	**SMD**	**95%CI SMD**	**k**
TG	NA	NA	2	**0.96**	**0.42–1.49**	4	0.32	−0.08–0.71	3
TC	NA	NA	2	**1.72**	**0.59–2.85**	4	−1.22	−2.61–0.17	3
LDL-C	NA	NA	2	**1.66**	**1.17–2.14**	4	NA	NA	2
HDL-C	NA	NA	2	**−0.72**	**−1.08 to −0.37**	4	NA	NA	2
VLDL-C	NA	NA	0	NA	NA	0	NA	NA	0
TG/HDL ratio	NA	NA	0	NA	NA	0	NA	NA	0
	**Australia and New Zealand**								
**Lipid profile parameter**	**1st trimester**			**2nd trimester**			**3rd trimester**		
	SMD	95%CI SMD	k	SMD	95%CI SMD	k	SMD	95%CI SMD	k
TG	0.33	−0.70–1.36	3	**0.62**	**0.16–1.09**	6	NA	NA	2
TC	NA	NA	1	−0.33	−0.85–0.18	3	NA	NA	1
LDL-C	NA	NA	0	NA	NA	0	NA	NA	1
HDL-C	NA	NA	1	**−0.50**	**−0.88 to −0.17**	3	NA	NA	1
VLDL-C	NA	NA	0	NA	NA	0	NA	NA	0
TG/HDL ratio	NA	NA	0	NA	NA	0	NA	NA	0

k—Number of included studies; NA—not applicable, Standardized Mean Difference (SMD), *p*-values that reached statistical significance are shown in bold.

**Table 5 medicina-61-01290-t005:** Meta-analysis of lipid profile parameters (TG, TC, LDL, HDL, VLDL, and TG/HDL ratio) in GDM compared to non-GDM pregnancies according to the study design.

Lipid Profile Parameter	Retrospective								
	1st Trimester			2nd Trimester			3rd Trimester		
	SMD	95%CI SMD	k	SMD	95%CI SMD	k	SMD	95%CI SMD	k
TG	**0.48**	**0.35–0.60**	31	**0.59**	**0.48–0.69**	75	**0.50**	**0.31–0.68**	30
TC	**0.31**	**0.20–0.42**	26	**0.37**	**0.22–0.51**	72	**0.30**	**0.02–0.58**	30
LDL-C	**0.27**	**0.13–0.41**	24	**0.18**	**0.06–0.30**	66	0.21	−0.03–0.45	24
HDL-C	**−0.15**	**−0.18 to −0.12**	28	**−0.44**	**−0.55 to −0.33**	68	−0.31	−0.76–0.14	23
VLDL-C	NA	NA	1	**1.57**	**0.65–2.49**	7	NA	NA	2
TG/HDL ratio	NA	NA	1	**0.52**	**0.19–0.86**	4	NA	NA	1
	**Prospective**								
**Lipid profile parameter**	**1st trimester**			**2nd trimester**			**3rd trimester**		
	**SMD**	**95%CI SMD**	** ^1^ ** **k**	**SMD**	**95%CI SMD**	**k**	**SMD**	**95%CI SMD**	**k**
TG	**0.64**	**0.49–0.80**	30	**0.58**	**0.45–0.71**	46	**0.37**	**0.23–0.50**	36
TC	**0.62**	**0.43–0.82**	27	**0.17**	**0.06–0.29**	43	−0.07	−0.18–0.05	33
LDL-C	**0.45**	**0.29–0.61**	26	0.13	−0.01–0.27	35	−0.09	−0.22–0.04	29
HDL-C	**−0.20**	**−0.24 to −0.17**	24	**−0.17**	**−0.31 to −0.04**	36	−0.13	−0.26–0.01	30
VLDL-C	NA	NA	0	NA	NA	1	0.21	−0.43–0.85	3
TG/HDL ratio	NA	NA	2	NA	NA	1	NA	NA	2
	**Cross-sectional**								
**Lipid profile parameter**	**1st trimester**			**2nd trimester**			**3rd trimester**		
	**SMD**	**95%CI SMD**	**k**	**SMD**	**95%CI SMD**	**k**	**SMD**	**95%CI SMD**	**k**
TG	**2.34**	**0.63–4.05**	6	**0.54**	**0.36–0.73**	33	**0.28**	**0.05–0.50**	12
TC	0.12	−0.93–1.16	6	0.20	−0.03–0.43	32	**−0.30**	**−050 to −0.09**	12
LDL-C	−0.27	−1.26–0.73	6	0.18	−0.04–0.39	31	**−0.22**	**−0.38 to −0.06**	12
HDL-C	0.84	−0.04–0.21	6	**−0.45**	**−0.61 to −0.29**	31	**−0.24**	**−0.36 to −0.13**	12
VLDL-C	NA	NA	1	**0.32**	**0.09–0.54**	3	NA	NA	2
TG/HDL ratio	NA	NA	1	**1.19**	**0.97–1.41**	4	NA	NA	0

k—Number of included studies; NA—not applicable, Standardized Mean Difference (SMD), *p*-values that reached statistical significance are shown in bold.

**Table 6 medicina-61-01290-t006:** Meta-analysis of lipid profile parameters (TG, TC, LDL, HDL, VLDL, and TG/HDL ratio) in GDM compared to non-GDM pregnancies according to the criteria used for GDM diagnosis.

Lipid Profile Parameter	IADPSG								
	1st Trimester			2nd Trimester			3rd Trimester		
	SMD	95%CI SMD	k	SMD	95%CI SMD	k	SMD	95%CI SMD	k
TG	**0.58**	**0.48–0.68**	42	**0.56**	**0.47–0.64**	80	**0.49**	**0.34–0.64**	42
TC	**0.33**	**0.23–0.43**	41	**0.22**	**0.11–0.32**	74	**3.29**	**3.08–3.50**	42
LDL-C	**0.33**	**0.21–0.44**	37	**0.18**	**0.10–0.26**	68	0.021	−0.13–0.17	39
HDL-C	**−0.37**	**−0.51 to −0.23**	37	**−0.29**	**−0.41 to −0.17**	71	−0.16	0.44–0.13	41
VLDL-C	NA	NA	1	**0.98**	**0.24–1.72**	5	0.22	−0.43–0.85	3
TG/HDL ratio	NA	NA	1	0.52	−0.13–1.17	3	NA	NA	0
	**ADA**								
**Lipid profile parameter**	**1st trimester**			**2nd trimester**			**3rd trimester**		
	**SMD**	**95%CI SMD**	** ^1^ ** **k**	**SMD**	**95%CI SMD**	**k**	**SMD**	**95%CI SMD**	**k**
TG	**1.07**	**0.53–1.62**	12	**0.54**	**0.41–0.68**	67	**0.51**	**0.18–0.83**	15
TC	**1.37**	**0.72–2.02**	9	**0.17**	**0.07–0.26**	63	**3.53**	**3.00–4.07**	15
LDL-C	**0.75**	**0.31–1.18**	9	**0.21**	**0.07–0.34**	61	0.15	−0.22–0.53	13
HDL-C	0.02	−0.35–0.37	10	**−0.25**	**−0.38 to −0.13**	62	−0.05	−0.33–0.23	12
VLDL-C	NA	NA	0	**1.57**	**0.43–2.70**	5	NA	NA	2
TG/HDL ratio	NA	NA	1	**0.69**	**0.21–1.16**	4	NA	NA	1
	**WHO**								
**Lipid profile parameter**	**1st trimester**			**2nd trimester**			**3rd trimester**		
	**SMD**	**95%CI SMD**	**k**	**SMD**	**95%CI SMD**	**k**	**SMD**	**95%CI SMD**	**k**
TG	**0.42**	**0.27–0.57**	6	**0.51**	**0.31–0.71**	21	**0.42**	**0.16–0.68**	6
TC	**0.24**	**0.09–0.39**	6	0.38	−0.09–0.85	21	**3.21**	**2.35–4.06**	5
LDL-C	0.05	−0.26–0.37	5	0.19	−0.04–0.42	21	0.34	−0.12–0.81	5
HDL-C	**−0.25**	**−0.44 to −0.06**	6	**−0.33**	**−0.53 to −0.13**	21	−0.39	−0.83–0.06	5
VLDL-C	NA	NA	0	NA	NA	2	NA	NA	0
TG/HDL ratio	NA	NA	0	NA	NA	1	NA	NA	0

k—Number of included studies; NA—not applicale, Standardized Mean Difference (SMD), *p*-values that reached statistical significance are shown in bold.

**Table 7 medicina-61-01290-t007:** Meta-analysis of lipid profile parameters (TG, TC, LDL, HDL, VLDL, and TG/HDL ratio) in GDM compared to non-GDM pregnancies according to smoking habit (smoker/non-smoker) and presence of obesity in the study groups.

Lipid Profile Parameter	Smokers and Non-Smokers								
	1st Trimester			2nd Trimester			3rd Trimester		
	SMD	95%CI SMD	k	SMD	95%CI SMD	k	SMD	95%CI SMD	k
TG	**0.37**	**0.22–0.51**	13	**0.50**	**0.30–0.70**	18	**0.25**	**0.10–0.41**	12
TC	**0.24**	**0.17–0.32**	11	**0.22**	**0.07–0.37**	16	0.05	−0.14–0.25	12
LDL-C	**0.27**	**0.14–0.39**	9	**0.37**	**0.15–0.59**	15	−0.11	−0.27–0.05	8
HDL-C	**−0.54**	**−0.81 to −0.28**	11	**−0.31**	**−0.52 to −0.11**	16	−0.74	−1.70–0.21	8
VLDL-C	NA	NA	2	NA	NA	0	NA	NA	2
TG/HDL ratio	NA	NA	0	NA	NA	0	NA	NA	0
	**Non-smokers exclusively**								
**Lipid profile parameter**	**1st trimester**			**2nd trimester**			**3rd trimester**		
	**SMD**	**95%CI SMD**	**k**	**SMD**	**95%CI SMD**	**k**	**SMD**	**95%CI SMD**	**k**
TG	**0.40**	**0.34–0.46**	13	**0.51**	**0.35–0.68**	33	**0.66**	**0.47–0.85**	24
TC	**0.44**	**0.17–0.71**	12	0.06	−0.08–0.19	33	0.22	−0.03–0.47	22
LDL-C	**0.51**	**0.13–0.78**	12	0.04	−0.16–0.23	30	−0.06	−0.23–0.11	24
HDL-C	**−0.17**	**−0.29 to −0.05**	12	**−0.36**	**−0.53 to −0.20**	30	−0.12	−0.34–0.10	24
VLDL-C	NA	NA	0	NA	NA	1	NA	NA	0
TG/HDL ratio	NA	NA	0	NA	NA	1	NA	NA	0
	**Obese and non-obese pregnancies included**								
**Lipid profile parameter**	**1st trimester**			**2nd trimester**			**3rd trimester**		
	**SMD**	**95%CI SMD**	**k**	**SMD**	**95%CI SMD**	**k**	**SMD**	**95%CI SMD**	**k**
TG	**0.62**	**0.25–0.98**	10	**0.57**	**0.42–0.72**	43	**0.72**	**0.36–1.08**	17
TC	**0.26**	**0.17–0.35**	10	**0.23**	**0.10–0.37**	40	−0.05	−0.46–0.36	16
LDL-C	−0.04	−0.32–0.24	7	0.11	−0.07–0.30	32	**0.42**	**0.01–0.83**	15
HDL-C	**−0.31**	**−0.04 to −0.22**	8	**−0.24**	**−0.45 to −0.04**	33	**−0.44**	**−0.87 to 0.02**	15
VLDL-C	NA	NA	1	**1.17**	**0.32–2.02**	4	NA	NA	2
TG/HDL ratio	NA	NA	1	NA	NA	1	NA	NA	2
	**Non-obese pregnancies exclusively**								
**Lipid profile parameter**	**1st trimester**			**2nd trimester**			**3rd trimester**		
	**SMD**	**95%CI SMD**	**k**	**SMD**	**95%CI SMD**	**k**	**SMD**	**95%CI SMD**	**k**
TG	**0.61**	**0.50–0.72**	62	**0.56**	**0.49–0.64**	177	**0.47**	**0.36–0.58**	77
TC	**0.47**	**0.35–0.59**	55	**0.20**	**0.12–0.28**	165	0.09	−0.04–0.21	72
LDL-C	**0.39**	**0.27–0.50**	52	**0.19**	**0.11–0.26**	155	0.01	−0.10–0.12	66
HDL-C	**−0.35**	**−0.49 to −0.22**	55	**−0.32**	**−0.40 to −0.25**	160	**−0.25**	**−0.41 to −0.10**	67
VLDL-C	NA	NA	2	**0.74**	**0.24–1.24**	11	0.27	−0.28–0.83	4
TG/HDL ratio	**0.43**	**0.28–0.59**	4	**0.71**	**0.35–1.06**	8	NA	NA	0

k—Number of included studies; NA—not applicale, Standardized Mean Difference (SMD), *p*-values that reached statistical significance are shown in bold.

## Data Availability

All additional data are available as Appendix A.

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
