# Peer review of "Trimester-Specific Serum Lipid Profiles in Gestational Diabetes Mellitus: A Systematic Review, Meta-Analysis, and Meta-Regression"

_medicina, 2025, doi:10.3390/medicina61071290_

Round 1
Reviewer 1 Report
Comments and Suggestions for Authors
Authors present a systematic review, meta-analysis, and meta-regression on the association between serum lipid parameters and gestational diabetes mellitus (GDM). The authors have synthesized a vast body of literature (457 studies for qualitative analysis, with a significant subset for quantitative analysis) to evaluate trimester-specific differences in triglycerides (TG), total cholesterol (TC), LDL-C, HDL-C, VLDL-C, and the TG/HDL ratio between pregnant women with and without GDM.
The manuscript is well-suited for publication but requires several revisions to enhance clarity, strengthen the interpretation of some findings, and ensure full data transparency.
Major comments:
- On page 5 (lines 231-232), the authors state: "Forest plots were constructed, but due to the size of the forest plot figures, they were not displayed." While the decision to exclude dozens of forest plots from the main text is understandable for readability, their complete omission is a significant limitation. Forest plots are an essential component of a meta-analysis, allowing readers to visualize heterogeneity, the effect size and confidence interval of each included study, and each study's relative weight in the pooled estimate. It is imperative that the full set of forest plots for every meta-analysis performed is made available and they can be added as Supplementary Material.
- In the "Strengths and Limitations" section of the Discussion, the authors should add a note of caution regarding the interpretation of the numerous subgroup analyses, acknowledging the inflated risk of false positives. They should focus their discussion on the most consistent and clinically meaningful patterns that emerge, rather than giving equal weight to every statistically significant p-value.
- The phrase "identity card" in the title is not very rigorous, I would give a more academic title “Trimester-Specific Serum Lipid Profiles in Gestational Diabetes Mellitus: A Systematic Review, Meta-Analysis, and Meta-Regression”
Minor comments:
- Abstract (line 30): The phrase "...in pregnant women with compared to pregnant women without GDM" is grammatically awkward. I suggest rephrasing to: "...in pregnant women with GDM compared to those without the condition."
- Results (page 7, lines 275-278):The description of study designs is slightly confusing as the percentages provided do not seem to sum correctly (33% + 20% + 11% + 27% + 14% > 100%). Please clarify this section by providing the exact numbers (N) and corresponding percentages for each category, ensuring the total accounts for all 457 studies. For example: "Of the 457 included studies, the design was retrospective in N (X%), prospective in N (Y%), and cross-sectional in N (Z%). The study design was not reported in N (A%) and was unclear in N (B%)."
- Results (Table 1, page 9):This is an excellent summary table. However, in the subsequent, more complex tables (e.g., Table 3, 4), the header ¹k or k is used for the number of studies. For absolute clarity and consistency, please define this abbreviation in the footnote of every table where it appears (e.g., "k = number of included studies").
- Discussion (Clinical Implications of Trimester-Specific Findings):The discussion is strong. It could be further enhanced by expanding on the specific clinical implications of the trimester-specific findings. For instance, why is it important that TC and LDL-C elevations are significant in the first and second trimesters but not the third? This directly relates to the optimal window for early screening and intervention, a key theme of the manuscript.
- Lines 232-233:The sentence "...instead, more informative and clearer tables were created" sounds slightly defensive. I suggest a more neutral and standard phrasing: "To clearly summarize the large volume of data, pooled results are presented in summary tables. The complete forest plots for each analysis are available in the Supplementary Materials."
Author Response
Dear reviewer, we are very grateful for your review of our research. In the following text, you will find our answers to your objections, as well as their corrections. If we did not respond to some objections by changing the manuscript, we gave a reason for it. We hope for your positive review after the corrections. Thank you.
Major comments:
Reviewer objection: On page 5 (lines 231-232), the authors state: "Forest plots were constructed, but due to the size of the forest plot figures, they were not displayed." While the decision to exclude dozens of forest plots from the main text is understandable for readability, their complete omission is a significant limitation. Forest plots are an essential component of a meta-analysis, allowing readers to visualize heterogeneity, the effect size and confidence interval of each included study, and each study's relative weight in the pooled estimate. It is imperative that the full set of forest plots for every meta-analysis performed is made available and they can be added as Supplementary Material.
Answer: We completely agree with the reviewer comment regarding non-omission of forest plots. We had given an effort to present results in most transparent way, thus we added forest plots, as well as funnel plots for publication bias assessment, for main outcomes as Supplement Materials.
Reviewer objection: In the "Strengths and Limitations" section of the Discussion, the authors should add a note of caution regarding the interpretation of the numerous subgroup analyses, acknowledging the inflated risk of false positives. They should focus their discussion on the most consistent and clinically meaningful patterns that emerge, rather than giving equal weight to every statistically significant p-value.
Answer: We added the Strengths and Limitations section first, in order to distinguish it from the main corpus of the discussion. Second, we added the sentence: An important limitation within analysis is the extensive number of subgroup analyses that raises concerns about inflated type I error rates, increasing the likelihood of false-positive findings; therefore, these results warrant cautious interpretation and independent replication. We have carefully considered your suggestion and revised the discussion so it ensured that the analysis highlights key findings with clinical relevance.
Reviewer objection: The phrase "identity card" in the title is not very rigorous, I would give a more academic title “Trimester-Specific Serum Lipid Profiles in Gestational Diabetes Mellitus: A Systematic Review, Meta-Analysis, and Meta-Regression”
Answer: We agree that the proposed title, “Trimester-Specific Serum Lipid Profiles in Gestational Diabetes Mellitus: A Systematic Review, Meta-Analysis, and Meta-Regression”, is more academically appropriate and reflective of the study's scope and methodology, thus we accepted the proposal.
Minor comments:
Reviewer objection: Abstract (line 30): The phrase "...in pregnant women with compared to pregnant women without GDM" is grammatically awkward. I suggest rephrasing to: "...in pregnant women with GDM compared to those without the condition."
Answer: We rephrased the sentence.
Reviewer objection: Results (page 7, lines 275-278): The description of study designs is slightly confusing as the percentages provided do not seem to sum correctly (33% + 20% + 11% + 27% + 14% > 100%). Please clarify this section by providing the exact numbers (N) and corresponding percentages for each category, ensuring the total accounts for all 457 studies. For example: "Of the 457 included studies, the design was retrospective in N (X%), prospective in N (Y%), and cross-sectional in N (Z%). The study design was not reported in N (A%) and was unclear in N (B%).
Answer: We clarify the section regarding study designs by adding exact numbers as recommended.
Reviewer objection: Results (Table 1, page 9): This is an excellent summary table. However, in the subsequent, more complex tables (e.g., Table 3, 4), the header ¹k or k is used for the number of studies. For absolute clarity and consistency, please define this abbreviation in the footnote of every table where it appears (e.g., "k = number of included studies").
Answer: We define the abbreviation in the footnote of every table where it appears.
Reviewer objection :The discussion is strong. It could be further enhanced by expanding on the specific clinical implications of the trimester-specific findings. For instance, why is it important that TC and LDL-C elevations are significant in the first and second trimesters but not the third? This directly relates to the optimal window for early screening and intervention, a key theme of the manuscript.
Answer: We touched on this proposal in the initial part of the discussion where we mentioned our trimester specific results.
Reviewer objection: Lines 232-233: The sentence "...instead, more informative and clearer tables were created" sounds slightly defensive. I suggest a more neutral and standard phrasing: "To clearly summarize the large volume of data, pooled results are presented in summary tables. The complete forest plots for each analysis are available in the Supplementary Materials."
Answer: We rephrased the sentence.
Reviewer 2 Report
Comments and Suggestions for Authors
Thank you for inviting me to review this manuscript. My specific concerns regarding the paper are included below:
- A native English speaker or a professional editing service should improve the language quality.
- Tg/HDL should consistently be written as TG/HDL throughout the manuscript.
- The search timeframe is not mentioned in the abstract.
- Some terms (i.e. SMD) are used without being spelt out at first mention. This reduces accessibility for broader audiences.
- The conclusion emphasizes decreased HDL-C, yet the results section does not report a statistically significant difference for HDL-C or state any SMD values for it. There is a logical mismatch between the findings and what is emphasized in the conclusion.
- Definitions of outcomes (i.e. measured levels) should specify whether outcomes were fasting or non-fasting, units, assay types, etc.
- No clarity on whether duplicate data from overlapping studies (i.e. multiple publications from the same cohort) were handled in the methods section.
- The exact search terms, date range, and language filters are not specified in the methods section.
- The use of both NOS and Jadad is conceptually inconsistent since Jadad is designed for RCTs, whereas most included studies appear to be observational.
- Kindly include quality scores as a supplementary table.
- The justification for using SMD rather than WMD is unclear. Were all lipids reported in different units or measured by varied techniques?
- I did not see the Forest plots. This is a crucial limitation. At a minimum, a representative plot should be included in the main or supplementary materials.
- Additionally, I did not see a publication bias assessment using funnel plots.
Author Response
Dear reviewer, we are very grateful for your review of our research. In the following text, you will find our answers to your objections, as well as their corrections. If we did not respond to some objections by changing the manuscript, we gave a reason for it. We hope for your positive review after the corrections. Thank you.
Reviewer objection: A native English speaker or a professional editing service should improve the language quality
Answer: Dear reviewer we hired a native English person to review the manuscript in detail and made certain changes to the text.
Reviewer objection Tg/HDL should consistently be written as TG/HDL throughout the manuscript
Answer: We changed it to TG/HDL. Also, we checked other abbreviations.
Reviewer objection: The search timeframe is not mentioned in the abstract.
Answer: We added the timeframe as recommended.
Reviewer objection: Some terms (i.e. SMD) are used without being spelt out at first mention. This reduces accessibility for broader audiences.
Answer: We checked and defined all necessary terms at first mention.
Reviewer objection The conclusion emphasizes decreased HDL-C, yet the results section does not report a statistically significant difference for HDL-C or state any SMD values for it. There is a logical mismatch between the findings and what is emphasized in the conclusion.
Answer: We conducted a thorough review and identified the missing components. The conclusion has been revised
Reviewer objection: Definitions of outcomes (i.e. measured levels) should specify whether outcomes were fasting or non-fasting, units, assay types, etc.
Answer: Units are added to Supplement material 6 for lipids (TG, TC, LDL-C, HDL-C, VLDL-C). The explanation regarding converting different units into unique mmol/l is added into the method. We used Standardized Mean Difference (SMD) as the measure of the overall effect size to cope with different methods for lipid parameters measurement, thus we did not extract explicitly every possible assay type.
Reviewer objection: No clarity on whether duplicate data from overlapping studies (i.e. multiple publications from the same cohort) were handled in the methods section.
Answer: It is of great importance to be clear how scientist cope with this situation. We added the explanation how we dealt with it into the Methodology section. We considered and included only the study with the larger sample size in cases where there was overlap in the examined cohort. Fortunately, although 457 studies were included in this systematic review, there were only a few extra of such instances.
Reviewer objection: The exact search terms, date range, and language filters are not specified in the methods section.
Answer: Search terms within search queries for all three electronic databases and date range are reported in Methodology section, Search strategy subsection. Previously it was in a form of text but now we decided to present those segments in a table in order to be more transparent and easier to read.
Reviewer objection: The use of both NOS and Jadad is conceptually inconsistent since Jadad is designed for RCTs, whereas most included studies appear to be observational.
Answer: It`s right, we used NOS explicitly for observational, and Jadad scale for interventional studies. We presented as recommended the table with quality assessment results as a Supplement material: Table S6.
Reviewer objection: Kindly include quality scores as a supplementary table.
Answer: We added Supplement material: Table S6, and a subsection Quality assessment within results.
Reviewer objection: The justification for using SMD rather than WMD is unclear. Were all lipids reported in different units or measured by varied techniques?
Answer: When different studies use different measurement scales for the same continuous outcome, like different assay types for lipid profile parameters in our situation as studies were published between 1980 until 2024, standardized mean difference (SMD) is justified. It standardizes the mean difference by dividing it by the standard deviation, which enables comparison of outcomes expressed in different units. Also, its goal is to compare effects across studies with varying degrees of variability. SMD expresses the effect size in units of standard deviation (e.g., Cohen’s d), making it easier to compare relative effects between studies with different variabilities.
Reviewer objection: I did not see the Forest plots. This is a crucial limitation. At a minimum, a representative plot should be included in the main or supplementary materials.
Answer: We added major forest plots for main meta-analysis of lipid profile parameters differences per trimester into Supplement material.
Reviewer objection: Additionally, I did not see a publication bias assessment using funnel plots.
Answer: We stated that Funnel plot was used for assessing publication bias and that funnel plots are not shown but can be given on request in Methodology section, but now we added all funnel plots for main outcomes as Supplement Materials.
Round 2
Reviewer 1 Report
Comments and Suggestions for Authors
The authors have successfully addressed all of my concerns in a comprehensive and thoughtful manner. The revised manuscript is now a robust, clear, and significant contribution to the field. I have no further reservations.
Reviewer 2 Report
Comments and Suggestions for Authors
Thank you for the revisions